# Real-Time LIDAR-Based Urban Road and Sidewalk Detection for Autonomous Vehicles

**DOI:** 10.3390/s22010194

**Published:** 2021-12-28

**Authors:** Ernő Horváth, Claudiu Pozna, Miklós Unger

**Affiliations:** 1Vehicle Industry Research Center, Széchenyi István University, H-9026 Győr, Hungary; pozna@sze.hu (C.P.); unger.miklos@ga.sze.hu (M.U.); 2Department of Product Design and Robotics, Transylvania University of Brasov, 500036 Brasov, Romania

**Keywords:** autonomous vehicle, open source, LIDAR point cloud, free-space detection, road segmentation, ground-non-ground segmentation, obstacle detection, autonomous vehicles, self-driving

## Abstract

Road and sidewalk detection in urban scenarios is a challenging task because of the road imperfections and high sensor data bandwidth. Traditional free space and ground filter algorithms are not sensitive enough for small height differences. Camera-based or sensor-fusion solutions are widely used to classify drivable road from sidewalk or pavement. A LIDAR sensor contains all the necessary information from which the feature extraction can be done. Therefore, this paper focuses on LIDAR-based feature extraction. For road and sidewalk detection, the current paper presents a real-time (20 Hz+) solution. This solution can also be used for local path planning. Sidewalk edge detection is the combination of three algorithms working parallelly. To validate the result, the de facto standard benchmark dataset, KITTI, was used alongside our measurements. The data and the source code to reproduce the results are shared publicly on our GitHub repository.

## 1. Introduction

Autonomous vehicles, also known as self-driving vehicles, are currently being rapidly improved. These vehicles or robots have the capability of sensing their environment and moving safely without human input. The four main tasks of these systems are sensing, perception, planning, and control. Sensing is basically data acquisition from the sensors; perception means feature extraction from the sensor data; planning means to create a feasible trajectory; finally, control is responsible for executing this trajectory. The scope of this work is perception, more precisely the extraction of road and sidewalk features. This information will be provided to the planning subsystem. From a sensor point of view, a self-driving vehicle incorporates vision-based methods, typically stereo or monocular cameras. Moreover, radio-waves-based ranging is present, most notably with radar. In the current work, our focus is on LIDAR, which is also a common light ranging sensor. While vision-based methods are suitable for many types of recognition, they are sensitive to illumination. LIDAR-based methods are largely invariant to illumination changes, and they are becoming inexpensive and more appropriate to measure the environment in 3D space in high resolution. This high resolution requires efficient algorithms to work in real time.

In an urban environment, curbs are meant to separate vehicle traffic from pedestrians. They are typically continuous along the road, and they isolate the elevated sidewalk level. LIDAR generates a discrete point cloud that provides 3D data encompassing height information. Thus, a simple approach would be to use height data and fit a plane to the road level for segmentation. This proves to be a naïve method for several reasons. The road is purposely designed to be curbed and not a perfectly flat surface. The drainage gradient is the effect of the combined road slope, the longitudinal slope, and surface cross-slope. Without this slope, the water would cover the surface, which leads to dangerous situations such as hydroplaning. Furthermore, over the years the concrete or the asphalt can accumulate numerous deformations. The unevenness caused by manholes, potholes, and others, which are common characteristics of an urban environment, makes feature extraction challenging. Even if there are some limitations regarding the lanes, the junctions, the minimum and maximum curvatures, this still leaves practically unlimited versions of road and sidewalk shapes. For LIDAR-based road and sidewalk detection, as a solution to these challenges, a real-time, robust, feasibly less parameter-sensitive algorithm was designed and implemented.

To present our method, the paper is organized as follows: Section 2 presents a review of the state of the art; Section 3 presents the proposed method; Section 4 describes the evaluation of the proposed method; and finally, Section 5 shows the conclusions.

## 2. Related Work

In the following, a review of related research is introduced. There are numerous papers on road and lane detection based on camera or camera and LIDAR fusion [1,2,3,4,5]. These solutions generally rely on image data and use mainly deep learning with neural networks. The neural networks are usually convolutional, and the models are trained using labelled images of urban roads and other scenarios. If the lightning conditions are satisfactory these solutions can perform in real time with GPU acceleration.

There are also solely LIDAR-based solutions that focus on ground detection [6,7] and road detection [8,9,10,11,12,13]. Most of them work in a postprocessing or semiautomated manner [8,9,10,11]. For LIDAR segmentation, there are neural-network-based solutions too, such as RangeNet++, PolarNet, etc. [14,15,16]. These solutions heavily rely on GPU. The current state of art ensures real-time performance with the help of FPGA or GPU acceleration [12,13,14,15]. Although numerous road shapes (T-shape, Y-shape, +-shape) have been investigated, most of the methods cannot handle complex intersections [8,9,10,11,12]. These shapes lack roundabouts, bridges, multilanes, safety islands, etc. LIDAR sensors provide raw 3D spatial information and can be organized as single-layer LIDAR (also referred as 2D LIDAR or laser scanner) and multilayer, or 3D LIDAR. In [17,18], a downward looking 2D LIDAR sensor scans the scene with 2D point array covering a 90° field-of-view. Their method extracts line segments from the raw data of the sensor in polar coordinates and the line segments are classified. [17] is a great example of this classical approach, which is also featured in the DARPA Urban Challenge. The limit in these solutions is the very narrow field of view compared to a 3D LIDAR. Fusion-based methods, which incorporate LIDAR/ultrasonic/camera/stereo camera [19,20,21,22], generally provide accurate outputs owing to the multiple sensor inputs. The limit of these techniques is their real-time performance: [20] mentions processing-time optimization as future work while [21] claims that ultrasonic fusion on an NVIDIA Xavier board is slower that the input; the method in [22] can work in real time with a high-end NVIDIA RTX 3090 GPU. Although lane detection is a related subject [23,24,25], different solutions may be applied for road map generation. These solutions aim for lane-level map generation instead of immediate lane keeping during autonomous driving. There are also solely 3D LIDAR-based solutions [26,27,28,29]. They usually work in a non-deterministic way, which is not necessarily a drawback. [27] uses random sample consensus (RANSAC) to filter for candidate point extraction and seed point extraction. [28] also uses RANSAC to estimate the quadratic polynomial model from candidate points and iterates continuously until the fitted model satisfies as many points as possible. [29] also uses heuristics; in order to estimate the ground plane, a sequential quadratic programming (SQP) optimization is applied. This solution also uses reflectance data and requires the sensor to be calibrated.

Some of the aforementioned methods have recognition issues. For example, in [8], determining the distance and angular difference between two adjacent points in the horizontal plane without elevation is challenging. In [9], determining the threshold of wavelet transformation can be also difficult, and may even provide false positive results. Moreover, in [10], trajectory is one of the inputs, which narrows down the usage range. Most of the methods discussed in [1,2,3,4,5,6,7,8,9,10,11,12,13,14,15,16,17,18,19,20,21,22,23,24,25,26,27,28,29,30,31,32,33] use LIDAR systems that spin at 10 Hz. This means a full 360° measurement is provided 10 times a second. The number of voxels can be calculated with respect to the horizontal angular resolution (typically 0.2°–0.8°, which means 1800–450 samples) and the vertical resolution (number of channels, typically 16–128). A 2D LIDAR has no vertical resolution; it consists of only one channel. The working range of the LIDAR is typically between 50 and 500 m. Our solution uses a 20 Hz frequency, 512 or 1028 vertical samples, and 64 horizontal channelled LIDAR, in 120 m range, and it is tested on multiple versions of available LIDARs (see the Section 4).

To apply road and sidewalk detection to our local planner, we expect generality, performance, real-time usage, lack of parameter sensitivity, and openness. Therefore, we decided to design and implement a solution that is based on a single sensor and offers robust results on roundabouts, multilanes, bridges, etc. The proposed method performs in real time without FPGA acceleration on a moderate embedded ARM CPU.

## 3. The Proposed Solution

The proposed solution finds the sidewalk using three different methods. It is important to mention that the output consists not only of the point clouds of the road and the separator area but also a simplified vector that is easily handleable. This output is useful for other algorithms, such as local path planners, because it is a more compact description of the road. Compactness can be one of the key components of real-time performance.

As a model of the urban road and sidewalk environment, a distorted flat surface and a slightly and unevenly elevated sidewalk can be imagined (see Figure 1). From a bird’s-eye view, the road and the sidewalk could take many forms. The mentioned properties and the simplified sensory data are illustrated in Figure 1. It is assumed that the LIDAR sensor is over the road surface level. The model of the urban road and sidewalk environment is displayed in Figure 1. The road is green, and the sidewalk is marked with red.

The proposed solution is free and available publicly as source code, called urban_road_filter. The input of our solution is a plain LIDAR stream, without a camera or any additional sensor data. The output is a 3D voxel point cloud of the road and sidewalk along with a 2D polygon description of the road. The solution encompasses three methods of sidewalk detection (star-shaped search method, X-zero, and Y-zero methods), a road detection method, and a 2D polygon-based road extraction.

### 3.1. Sidewalk Detection

Sidewalk edge detection is the combination of star-shaped search, X-zero, and Z-zero methods. All methods have the same purpose, but they work on a different principle. The final result is a logical disjunction of these methods’ outputs. It is important to note that the mentioned methods run in a parallel manner. False positive curb points may appear behind the actual curb. Curb points are the boundary voxels between the curb and the road. False positive curb points are created behind the curb, e.g., due to the similar 3D characteristics of the voxels from various artifacts. The artifact can be for example a public bench that stands out from the sidewalk the same way as the curb stands out from the road. This results in false identification. The final polygon is created between the road and first curb points, which means later curb points do not compromise the final result. This phenomenon does not affect the method negatively, because the false positive voxels are never on the road surface.

#### 3.1.1. Star-Shaped Search Method

This method divides the point cloud into rectangular-shaped segments. The combination of these shapes resembles a star; this is where the name originates. From each segment the possible sidewalk starting points are extracted, where the algorithm was created to not be sensitive to the Z-coordinate-based height variations. This means in practice that this algorithm would perform well even when a LIDAR is tilted with respect to the road surface plane. The point cloud is treated in the cylindrical coordinate system (see Figure 2).

Figure 3 represents the cut-out box (a rectangular cuboid) of the scanned points. The vertices of the box are represented by 8 points P_1,1; …; 2,4_. Its orientation and positioning change iteratively with incremental rotation and translation. More precisely, for each *β_k_* and k = 1, …, n_k_ rotations, the box is translated along the Δ direction by n_i_ successive increments. For an easier understanding of the proposed algorithm, Figure 3 illustrates the plane of symmetry π used in Figure 4. Figure 4 shows a side view of the cut-out box, i.e., the projection on plane π of the 2 boxes (B_k,2_, B_k,3_) and also the scanned points that are selected by the mentioned 2 boxes.

The proposed algorithm (see Figure 3, Figure 4 and Figure 5) contain the following steps:


*Step 1. The current box definition:*


The box is defined by 8 points $P_{1,1,},\ldots ,\;P_{2,4,}$. The geometric model of the iterated box, which is rotated with angle *β_k_* around the *Z* axis and translated with the *c_i_* value on the Δ axis (the box’s X axis), is:(1)$$P^{k,i}=R_{Z}\left(\beta_{k}\right)\cdot D_{X}\left(c_{i}\right)P^{0}$$
where the 8 points coordinates in their local coordinate system are:(2)$$P^{0}=\left[\begin{array}{cccc} x_{1,1}^{0} & x_{1,2}^{0} & \ldots & x_{2,4}^{0}\\ y_{1,1}^{0} & y_{1,2}^{0} & \ldots & y_{2,4}^{0}\\ z_{1,1}^{0} & z_{1,2}^{0} & \ldots & z_{2,4}^{0}\\ 1 & 1 & \ldots & 1 \end{array}\right]$$

The homogeneous translation transformed matrix is:(3)$$D_{X}\left(c_{i}\right)=\left[\begin{array}{cccc} 1 & 0 & 0 & c_{i}\\ 0 & 1 & 0 & 0\\ 0 & 0 & 1 & 0\\ 0 & 0 & 0 & 1 \end{array}\right]$$

The homogeneous rotation transformed matrix is:(4)$$D_{X}\left(c_{i}\right)=\left[\begin{array}{cccc} \cos(\beta_{k}) & -\sin(\beta_{k}) & 0 & 0\\ \sin(\beta_{k}) & \cos(\beta_{k}) & 0 & 0\\ 0 & 0 & 1 & 0\\ 0 & 0 & 0 & 1 \end{array}\right]$$

The coordinates of the 8 points in the global coordinate system, calculated after the rotation around the Z axes and translation on the Δ axes, are:(5)$$P^{k,i}=\left[\begin{array}{cccc} x_{1,1}^{k,i} & x_{1,2}^{k,i} & \ldots & x_{2,4}^{k,i}\\ y_{1,1}^{k,i} & y_{1,2}^{k,i} & \ldots & y_{2,4}^{k,i}\\ z_{1,1}^{k,i} & z_{1,2}^{k,i} & \ldots & z_{2,4}^{k,i}\\ 1 & 1 & \ldots & 1 \end{array}\right]$$
(6)$$\begin{array}{cc} c_{i}=i\cdot \Delta_{T} & \beta_{k}=k\cdot \Delta_{R} \end{array},\;\begin{array}{cc} i=1,\ldots ,n_{i} & k=1,\ldots ,n_{k} \end{array}$$
where $\Delta_{T}$ is the translation; $\Delta_{R}$ is the rotation increment; Δθ is the LIDAR angular increment, an interval parameter.


*Step 2. Inside points identification:*


The set of scanned points set $p=\left[\begin{array}{c} x\\ y\\ z \end{array}\right]$ witch are inside the *B_k,i_* box are defined like:(7)$$S_{k,i}=\left\{p|I\left(p,B_{k,i}\right)=1\right\}$$
where the function to determine if a point is inside the *B_k,i_* box is:(8)$$I\left(p,B_{k,i}\right)=\left\{\begin{array}{c} 1\;if\;V=\frac{1}{6}\overset{6}{\underset{r=1}{\sum }}F_{r}\left(p,B_{k,i}\right)\\ \;0\;else \end{array}\right.$$

The ensemble of subfunctions used in determining if the points are inside the box is:(9)$$\begin{array}{c} \left\{\begin{array}{c} F_{1}\left(p,B_{k,i}\right)=\begin{array}{c} \left|\det\left(\left[\begin{array}{ccc} P_{1,1}^{k,i}-p & P_{1,2}^{k,i}-p & P_{1,3}^{k,i}-p \end{array}\right]\right)\right|+\\ \left|\det\left(\left[\begin{array}{ccc} P_{1,3}^{k,i}-p & P_{1,4}^{k,i}-p & P_{1,1}^{k,i}-p \end{array}\right]\right)\right| \end{array}\\ F_{2}\left(p,B_{k,i}\right)=\begin{array}{c} \left|\det\left(\left[\begin{array}{ccc} P_{1,1}^{k,i}-p & P_{1,2}^{k,i}-p & P_{2,2}^{k,i}-p \end{array}\right]\right)\right|+\\ \left|\det\left(\left[\begin{array}{ccc} P_{2,2}^{k,i}-p & P_{2,4}^{k,i}-p & P_{1,1}^{k,i}-p \end{array}\right]\right)\right| \end{array}\\ F_{6}\left(p,B_{k,i}\right)=\begin{array}{c} \left|\det\left(\left[\begin{array}{ccc} P_{2,1}^{k,i}-p & P_{2,2}^{k,i}-p & P_{2,3}^{k,i}-p \end{array}\right]\right)\right|+\\ \left|\det\left(\left[\begin{array}{ccc} P_{2,3}^{k,i}-p & P_{2,4}^{k,i}-p & P_{2,1}^{k,i}-p \end{array}\right]\right)\right| \end{array} \end{array}\right.\\ P_{1,1}^{k,i}=\left[\begin{array}{c} x_{1,1}^{k,i}\\ y_{1,1}^{k,i}\\ z_{1,1}^{k,i} \end{array}\right] \end{array}$$

V=LlH; *L, l, H* the boxes dimension


*Step 3. The ordered set of the inside points computation:*


The ordered set has the same elements as $S_{k,i}$ but also characterized by the distance *ρ*
(10)$${\overset{-}{S}}_{k,i}=\left\{p_{k,i,j}|p_{k,i,j}\in S_{k,i}\wedge \rho_{k,i,j-1}\leq \rho_{k,i,j}\leq \rho_{k,i,j+1}\right\}$$
where $\rho_{k,i,j}$ is the distance of the scanned point $p_{k,i,j}$ on the Z axis, $p=\left[\begin{array}{c} x\\ y\\ z \end{array}\right]$, and $\rho =\sqrt{x^{2}+y^{2}}$


*Step 4. The relative high computation:*


The set of the relative highs (the distance on the Z axis computed for the points that are included in the ordered set)
(11)$$D_{k,i,j}=\left\{d_{k,i,j}|\begin{array}{c} d_{k,i,j}=abs\left(z_{k,i,j+1}-z_{k,i,j}\right)\wedge p_{k,i,j}\in {\overset{-}{S}}_{k,i}\wedge p_{k,i,j}=\\ {\left[\begin{array}{ccc} x_{k,i,j} & y_{k,i,j} & z_{k,i,j} \end{array}\right]}^{T} \end{array}\right\}$$
(12)$${\overset{-}{d}}_{k,i}=\frac{1}{m_{k.i}}\overset{m_{k,i}}{\underset{j=1}{\sum }}d_{k,i,j}$$
is the average distance between the ordered points inside the box
$$m_{k,i}=card\left(D_{k,i}\right)$$


*Step 5. Extracting the points of interest:*

(13)
$$\Gamma_{S}=\left\{p_{k,i,j}|abs\left({\overset{-}{d}}_{k,i}-d_{k,i,j}\right)\geq \varepsilon \right\}$$



The algorithm is represented in the block diagram of Figure 5.

**Figure 5 sensors-22-00194-f005:**
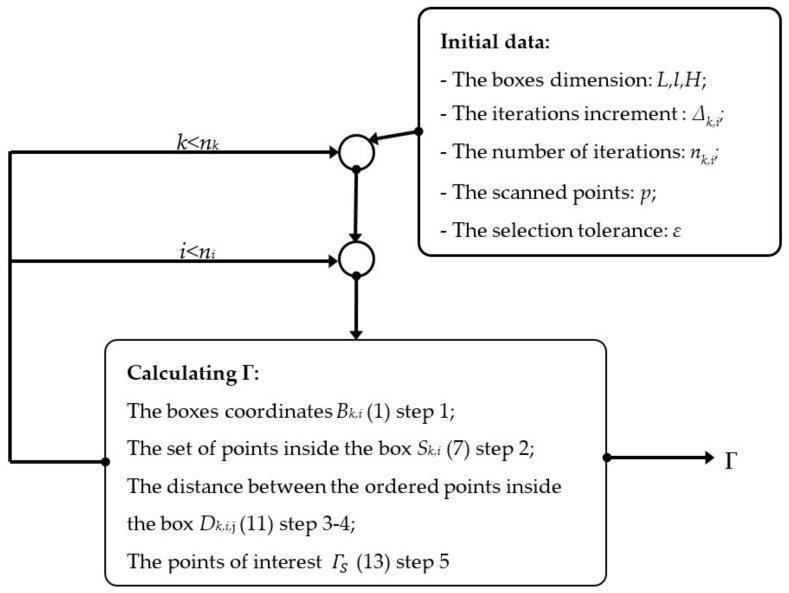
The proposed algorithm (star-shaped search method).

The points of interest represent the inner edge of the sidewalk. As previously mentioned, in order to increase the robustness of the method, we combined this output with the results computed with the X-zero and Z-zero.

#### 3.1.2. X-Zero Method

The X-zero and Z-zero methods find the sidewalk eluding the X and the Z component of the measurements. Both X-zero and Z-zero methods take the channel number of a voxel into account, thus it is necessary for the LIDAR not to be parallel with respect to the road surface plane. This is a known limitation of both mentioned algorithms, and thus of the whole urban_road_filter method.


*Step 1. Creation of the initial point set:*


The X-zero method omits the X dimension and uses cylindrical coordinates instead see Figure 6. The approach uses iterations through a ring (channel) and defines triangles on the voxels ($z_{i-1},z_{i},z_{i+1}$). The initial data are:(14)$$H=\left\{z_{i}|i=1,\ldots ,n\right\}$$
where *z_i_* is the height of the measured point $P_{i}$.


*Step 2. Sliding window definition:*


The method uses a sliding window ($W_{i}$) in an iterative manner, see Figure 7. The window is defined as follows: (15)$$W_{i}=\left\{z_{i-1},z_{i},z_{i+1}|z_{i,\ldots ,i+1}\in H,i=2,n-1\right\}$$

The point $P_{i}$’s angular parameter is defined as:(16)$$A=\left\{\alpha_{i}=f\left(z_{i-1},z_{i},z_{i+1}\right)|z_{i-1,i,i+1}\in W_{i}\wedge \left(z_{i-1}\langle z_{i}\rangle z_{i+1}\right),i=2,n-1\right\}$$
where:$$f\left(z_{i-1},z_{i},z_{i+1}\right)=\cos^{-1}\left(\frac{l_{i-1,i}^{2}+l_{i,i+1}^{2}-l_{i-1,i+1}^{2}}{2l_{i-1,i}l_{i,i+1}}\right)\;\left\{\begin{array}{c} l_{i-1,i}=\sqrt{{\left(z_{i}-z_{i-1}\right)}^{2}+\Delta^{2}}\\ l_{i,i+1}=\sqrt{{\left(z_{i}-z_{i+1}\right)}^{2}+\Delta^{2}}\\ l_{i-1,i+1}=\sqrt{{\left(z_{i+1}-z_{i-1}\right)}^{2}+4\Delta^{2}} \end{array}\right.$$
where Δ=r⋅Δθ is the linear increment; Δθ is the LIDAR angular increment, an interval parameter; r, is radius of the virtual cylinder on the surface on which are projected the measure points, *r* is a parameter chosen, after practical trails.


*Step 3. Extract the feature points:*


A set $\Gamma_{X}=\left\{\alpha |\alpha \in A\wedge \alpha <\alpha_{r}\right\}$ is the currently chosen data. f defines an angle and if it is bigger than the cylinder_deg_x (α_r_) parameter, it is considered as a high point. Please note that the parameters are listed in Section 3.3. The high point classification means that the voxel is considered as a curb feature. 

#### 3.1.3. Z-Zero Method

The Z-zero method is loosely inspired by the spatial feature extraction proposed in [12]. The main difference in our method is that a sliding-window-based approach (5 + 5 voxels by default) calculates the angles as a vector direction.


*Step 1. Initial data for the feature extraction:*


According to Figure 1 terminology, the angle $\alpha^{z0}$ can be defined as the angle between two vectors in the sliding window. Here, x and y are the coordinates of the points and n means the value specified as a parameter, which indicates the number of points on the curb side. In supplement to the angles defined by the vector, the change of height is considered to minimize false positive results. This is achieved by determining the one with the largest altitude value of both vector points and comparing it with the starting point of the vectors. If this meets the conditions, it is finally accepted as a sidewalk point. In the following (Equation (17)), n means the total number of points, x, y the dimensions, $v_{a}$ the road surface approximated direction, and $v_{b}$ the curb-approximated direction.
(17)$$v_{a}=\frac{1}{n}*\left[\overset{n}{\underset{k=1}{\sum }}\left(x_{i-k}-x_{i}\right),\overset{n}{\underset{k=1}{\sum }}\left(y_{i-k}-y_{i}\right)\right]\;v_{b}=\frac{1}{n}*\left[\overset{n}{\underset{k=1}{\sum }}\left(x_{i+k}-x_{i}\right),\overset{n}{\underset{k=1}{\sum }}\left(y_{i+k}-y_{i}\right)\right]\;\alpha_{1}^{z0}=\cos^{-1}\left(\frac{v_{a}*v_{b}}{\left|v_{a}\right|*\left|v_{b}\right|}\right)$$


*Step 2. Horizontal and vertical continuity search:*


As mentioned in [12], the Z-zero curb searching also relies on the definition of horizontal continuity, vertical continuity, angle threshold, and elevation threshold. Horizontal and vertical continuity examine whether the voxel has a large vertical or horizontal gap. If the gap is big enough it is considered as a candidate curb voxel. This behaviour can be adjusted by the curb_height parameter, listed in Section 3.3.


*Step 3. Angle and elevation search:*


Considering the two vectors in the sliding window, if a road surface voxel is examined, the angles between these should be around 180° (see Figure 1, $a_{2}$ angle). A threshold angle comparison (defined by the cylinder_deg_z parameter, listed in Section 3.3) decides whether the voxel is a curb point or not. Moreover, by default, a 5 cm elevation distance is acceptable between the neighbouring voxels. 


*Step 4. Extract the feature points:*


Similar to the X-zero method, the high point classification means that the voxel is considered as a curb feature. This means that the list of curb voxels is added to the possible curb voxel array, $\Gamma_{Z}$.

As a final step, the disjunct union of $\Gamma_{S}$ (from the star-shaped method), $\Gamma_{x}$ (from the X-zero method), and $\Gamma_{Z}$ (from the Y-zero method) is formed. This may encounter some false positive voxels, but the location of these never affects polygon-based representation.

### 3.2. Two-Dimensional Polygon-Based Road Representation

Along with the detection of the sidewalks, our algorithm also provides a polygon, i.e., a vector output of the detected road. This is created to be in between the road voxels and curb voxels. This output can be directly used by a local planner to design a feasible local trajectory. 

More precisely, the algorithm distinguishes two categories of road boundaries: the sidewalk including the obstacle-surrounded boundary (marked with a red stripe on Figure 8) and the out-of-range boundary (marked with a green stripe on Figure 8). This comes handy for a local planner who can process this kind of data. However, the local planner expects to know what is a hard boundary that no trajectory should cross. However, if a boundary appears due to out-of-range sensor data, a local planner can plan the trajectory in that direction. To reduce the boundary polygon quantity Douglas and Peucker’s [30,31] algorithm was used.

### 3.3. Parameter Settings

There are several parameters that may be used to fine-tune the solution, although even the default values produce adequate results. In the following, the parameters listed in Table 1 are introduced. An important parameter is the LIDAR topic and its frame name. It is important to know that the algorithm works with multiple methods at the same time. The size of the examined area can be set with multiple parameters. The region of interest (ROI) can be set by the x_direction parameter and with the minimum and maximum x, y, and z parameters. The x_direction parameter may have three different values: negative, positive, and both, indicating whether the region of interest is behind, in front of, or bidirectional related to the LIDAR along the x axis. The parameters called cylinder_deg_x, cylinder_deg_z, and sector_deg are the parameters in degrees for the X-zero, Z-zero, and star-shaped methods, respectively. These parameters can be set in a range of 0–180 degrees, and they were introduced in Section 3.1.2 and Section 3.1.3. The star-shaped method uses adaptive edge detection to find the points belonging to the curb. To tune this method, the kdev dispersion and kdiv distance coefficients need to be set. If the algorithm is used in a road where there are many potholes or failures, these parameters should be small. Correspondingly, if it is used in a high-quality road the kdev and kdiv parameters should be high.

## 4. Results

To evaluate the proposed method, broad real-time and log-file-based analysis and experiments were carried out. Our real-time tests were carried out at the ZalaZONE proving ground [32,33]. This facility has an extensive Smart City test field, which is designed to have the most significant properties of a traditional urban area. In this area, the dimensional drawings of the facility and drone images are available. With this additional information, we were able to elaborate our results even more accurately.

In Figure 9, three images are presented to explain our results more visually. The first image shows the road with green voxels and the sidewalk with red voxels. Although the false positive sidewalk points are visible, they do not affect the overall performance. The results were collected at 20 Hz and a 30 km/h speed. In addition, the exact RTK GPS location is associated with the LIDAR data, to get a more understandable result. The second image in Figure 9 shows an overlay of the drone image and our results, while the third image only shows the proving ground from above.

We collected a vast dataset at our university campus; in-vehicle tests were carried out there too (Figure 10). A part of these data is publicly available at the same repository as the code. The in-vehicle test was carried out on a single embedded controller (NVIDIA Jetson AGX Xavier) and the performance of the method was according to our expectations. The algorithm can perform at 20 Hz, which is the same frequency at which the LIDAR sensor provides its raw data. The Jetson AGX Xavier has an eight-core ARM architecture, 64-bit Carmel CPU which has a 2.26 GHz clock speed. During the in-vehicle tests, we used the Ubuntu 18.04 operating system, which is a common Linux distribution. During the log-file-based analysis, we observed that the solution ran up to 2× faster than real time, which meant 40 Hz in our case. In this case, an Intel Core i9 16-core x64 architecture i9-7960X CPU was used at a 2.80 GHz clock speed. During the log-file-based analysis, a Windows 11 machine with WSL was used.

To obtain a qualitative estimation of our method, we used [12] as a basis for the extensive comparison of similar algorithms. As an example of outcome, our results (left) compared with the results presented in [7] are displayed in Figure 11. 

We used another metric presented in [26] to quantify our results. To evaluate the algorithm, a high-end processor i9-77940X at 3.10 GHz was used and 16 GB RAM. On this architecture, the core of the algorithms, without the data acquisition and publication, was able to run 0.014 s on average, which means more than 70 Hz (see Figure 12). On a slightly slower CPU (i9-7960X at 2.80 GHz) but together with the data acquisition and publishing, it was still over 40 Hz. We calculated the running time of the algorithm proposed in the second part of the experiment, as shown in Figure 12. To compare our results to state-of-the art performance, in [26] the average running time was 0.036 s, this result was achieved on an older CPU (i7-4790) and the results were together with the data acquisition step. We are confident to say that both [26] and our method can perform in real-time.

To evaluate the proposed method and verify if it can meet the real-time requirements, we conducted quantitative experiments. For this purpose, we chose the ZalaZONE proving ground [32,33], which has urban roads without traffic. Figure 12 shows an example scenario where we drove for 75 s and recorded the LIDAR data at 20 Hz. We used the Ouster OS1 64 channel LIDAR with a range of 120 m. The LIDAR was at the top of the vehicle, 1.54 m above the ground without tilting. Our experiments validated that the proposed method can perform in real time both on a PC with an i9 CPU and an NVIDIA AGX Xavier embedded system. In this experiment, we used an RTK-INS (Novatel PwrPak7D GPS), which provides accurate position data at 20 Hz. 

## 5. Conclusions and Future Work

At the beginning of this project, our goal was to identify roads and curbs from LIDAR data. After the literature review, it turned out that there was no complete fit for our purposes. After extensive requirements listing, the work was elaborated by the authors and the research assistants listed in the acknowledgement section. The paper introduced a novel method for road and sidewalk detection. The sidewalk edge detection is the combination of the introduced star-shaped search, X-zero, and Z-zero methods in 3D voxels. Moreover, a polygon output was provided by the method, which can be directly used by a local trajectory planner. We evaluated the method via extensive real-time field tests and off-line analysis from previous measurements and public datasets. We compared the results of our solution to previous ones. 

The solution has limitations. Both the X-zero and the Z-zero algorithms require the LIDAR to be in a parallel position with respect to the road surface. Although this is a common sensor setup [34,35,36,37,38,39] and our vehicles were equipped this way, there are notable cases where it is recommended to set it up differently. For instance, [38] has LIDAR systems both straight (parallel to the road) and tilted. Furthermore, [40] only uses a tilted LIDAR, so as an example, our solution is unsuitable for this sensor setup. 

A new type of LIDAR, called solid-state technology is getting higher interest in the scientific community [41,42]. Although these sensors have not yet been fully commercialized, they promise a higher operational life and low power. They also produce structured 3D information but are organized differently. As a further limitation, the proposed method does not support this technology. The 3D data from the solid-state technology are not in the classical channel-based organization, thus our method cannot calculate the channel-based sliding window, on which the later steps rely. As mentioned, there are several limitations regarding our method. In addition, concatenating two traditional spinning LIDAR data produces a similar result. Concatenating multiple LIDAR streams is quite common, thus this is a more serious limitation. The concatenated LIDAR data from the method point of view resemble the solid-state data. The current method is not able to identify classic channel information.

Our approach is designed for autonomous vehicles; as future work, it could be extended to environmental monitoring [43] or map generation [27]. Additional future work may consist of overcoming the mentioned limitations. Similar to most software, the source code is subject to future developments. Thus, we have shared the necessary documentation, guidelines, and code of conduct. This also means that the code, dataset to test, and videos to understand are shared publicly at https://github.com/jkk-research/urban_road_filter (accessed on 22 December 2021).

## Figures and Tables

**Figure 1 sensors-22-00194-f001:**
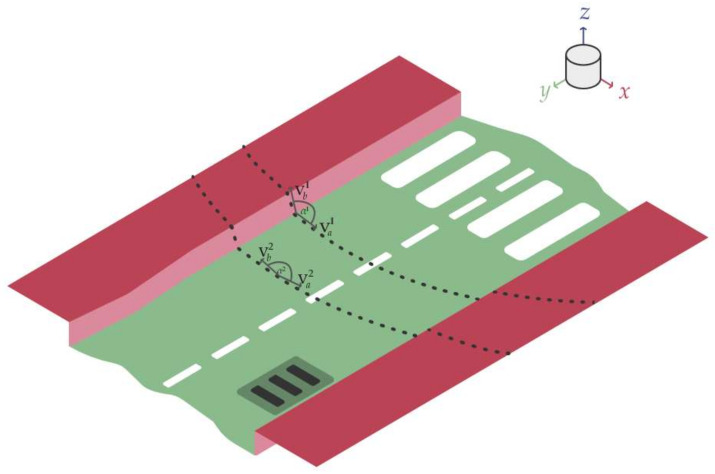
The figure displays the problem. The road is green, and the sidewalk is marked with red. Two channelled measurements are displayed as dotted lines. Moreover, some artifacts such as manholes and other unevenness are displayed.

**Figure 2 sensors-22-00194-f002:**
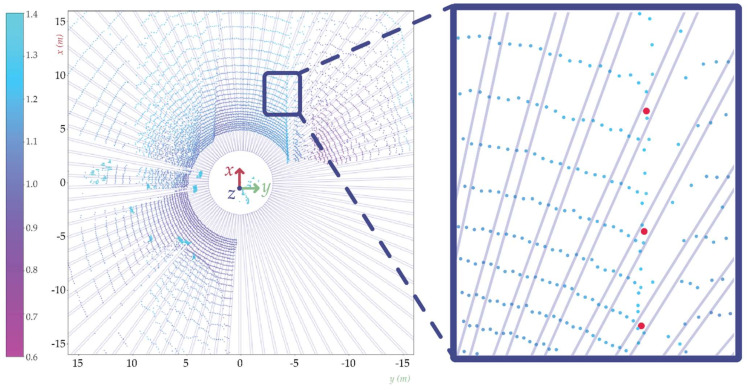
Star-shaped search method, the long rectangles (boxes) in circular layout represent the cut out from the original LIDAR point cloud. On the right in the zoomed image, the red dots are the sidewalk starting points.

**Figure 3 sensors-22-00194-f003:**
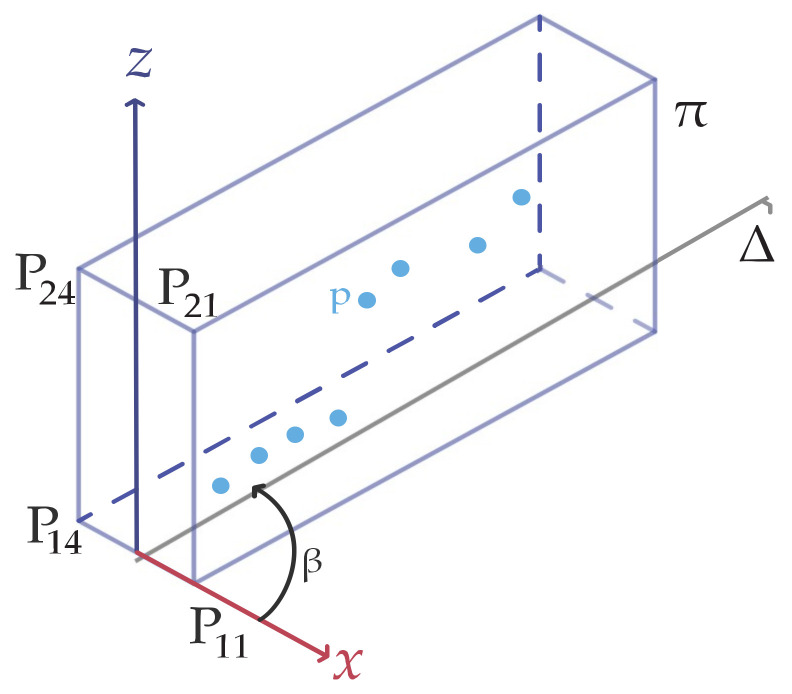
A single rectangle cut out of the scanned points.

**Figure 4 sensors-22-00194-f004:**
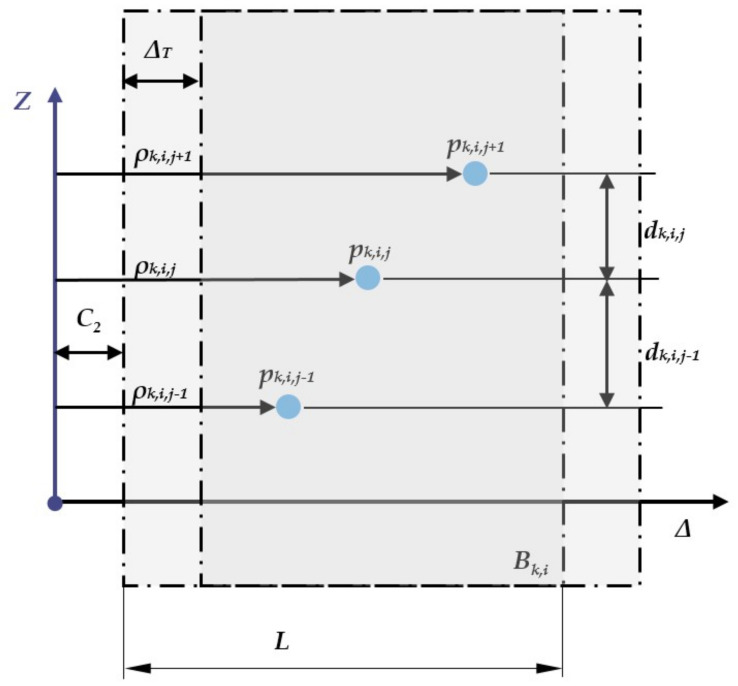
The graph’s side view concerning the points separation process and the *p_k,i,j_* points parameter selected by the *B_k,i_* box.

**Figure 6 sensors-22-00194-f006:**
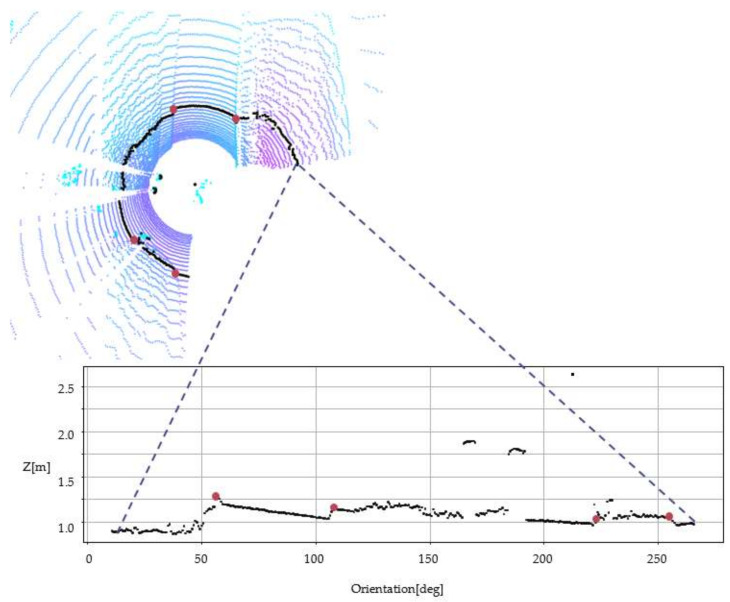
X-zero method, cylindrical coordinate system, single channel (ring).

**Figure 7 sensors-22-00194-f007:**
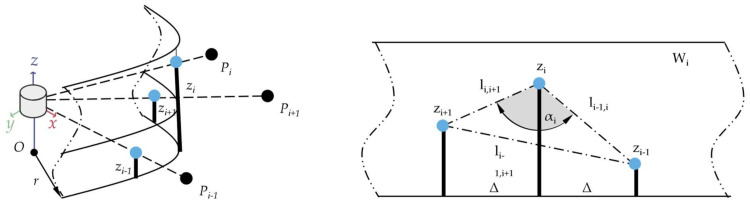
X-zero method, visualization of voxel triangles.

**Figure 8 sensors-22-00194-f008:**
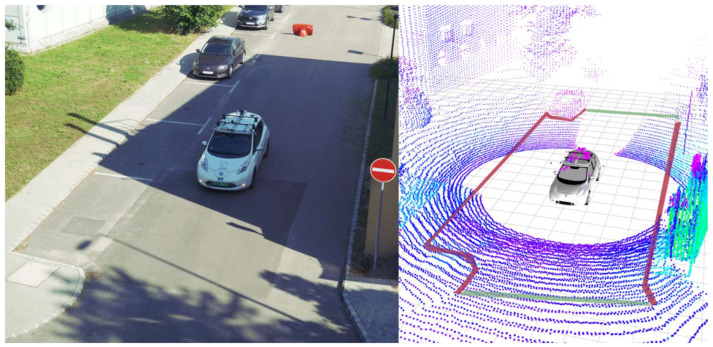
An example scenario explaining the 2D polygon road. The left image is not involved in the algorithm, it just helps to understand the first half of the scene better.

**Figure 9 sensors-22-00194-f009:**
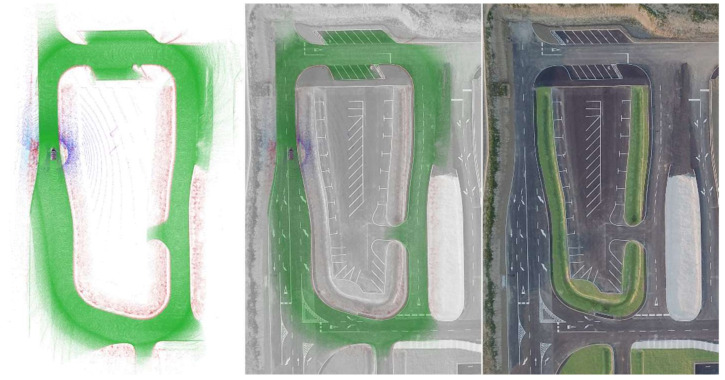
On the left side, the result is visible: road (green) and the sidewalk (red). In the middle, a mixture of the measurement and a drone image is displayed. On the right only the drone image is visible (location: Zalaegerszeg, Hungary, ZalaZONE).

**Figure 10 sensors-22-00194-f010:**
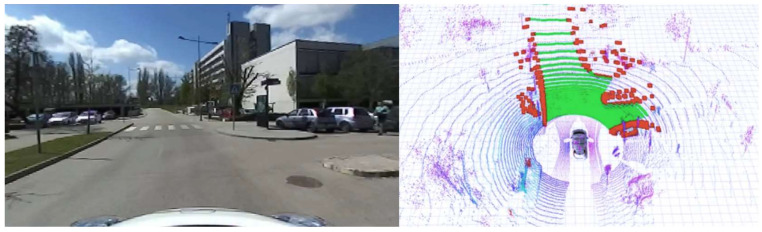
In-vehicle test of the method. On the (**left**) side, camera information is shown. On the (**right**) side, LIDAR 3D data are visible where the road (green) and sidewalk (red) are highlighted. The voxel scale is LIDAR intensity-based (location: University Campus, Győr, Hungary).

**Figure 11 sensors-22-00194-f011:**
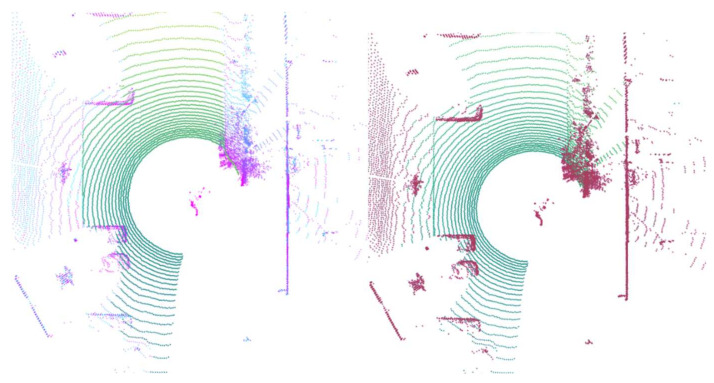
Results comparing our method (**left**) with the plane_fit_ground_filter (**right**) [7].

**Figure 12 sensors-22-00194-f012:**
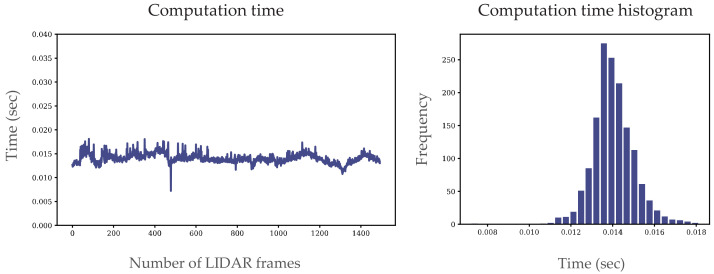
Results of the proposed approach (computation time).

**Table 1 sensors-22-00194-t001:** The parameter list of the urban road filter.

Param Name	Function	Type	(Interval)/Default
fixed_frame	The fixed frame from the transform list in ROS.	String	String
topic_name	The name of the LIDAR topic.	String	String
x_zero_method	A flag indicating whether the X-zero method is enabled.	Bool	(True-False)/True
z_zero_method	A flag indicating whether the Z-zero method is enabled.	Bool	(True-False)/True
star_shaped_method	A flag indicating whether the star-shaped method is enabled.	Bool	(True-False)/True
blind_spots	Filtering blind spots.	Bool	(True-False)/True
x_direction	Filtering x direction. Positive means in front of the LIDAR.	Both/ positive/negative	Both
interval	LIDAR’s vertical resolution.	Double	(0–10)/0.18
curb_height	Estimated minimum height of the curb (m).	Double	(0–10)/0.05
curb_points	Estimated number of points on the curb (pcs).	Int	(1–30)/5
beam_zone	Width of the beam zone (deg).	Double	(10–100)/30
cylinder_deg_x	The included angle of the examined triangle (three points) (deg) in x_zero_method.	Double	(0–180)/150
cylinder_deg_z	The included angle of the examined triangle (two vectors) (deg) in z_zero_method.	Double	(0–180)/140
sector_deg	Radial threshold (deg) in star_shaped_method.	Double	(0–180)/50
min_x,max_x,min_y,max_y,min_z,max_z	Size of the examined area x, y, z (m).	Double	(−200–200)/30
dmin_param	Minimum number of points for dispersion.	Int	(3–30)/10
kdev_param	Dispersion coefficient	Double	(0.5–5)/1.1225
kdist_param	Distance coefficient.	Double	(0.4–10)/2

## Data Availability

Source code and data available on https://github.com/jkk-research/urban_road_filter (accessed on 22 December 2021).

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
