# Peer review of "Real-Time LIDAR-Based Urban Road and Sidewalk Detection for Autonomous Vehicles"

_sensors, 2021, doi:10.3390/s22010194_

Round 1
Reviewer 1 Report
1.Line 58: The number of references is less than 30. Please add more description of related methods.
2.Line 68: Is it necessary to describe the simplified vector in details to make readers know what it is like?
3.Line 86: A comma misses before 'a road'.
4.Line 92: Are you trying to express that 'False positive curb points may appear ...' instead of 'May false positive curb points appear ...'? And please explain what false positive curb points are and why false positive voxels are never on the road surface.
5.Line 101: Please add the coordinate system in Figure 2.
6.Line 353: Is any data contained in the result of qualitative estimation? If so, please add a table to present the data like detection success rate and so on.
Author Response
Thank you for your review, we all agreed with your legitimate request and some of them were mentioned by other reviewers too, what also emphasizes how important they are.
1.Line 58: The number of references is less than 30. Please add more description of related methods.
Thank you, the reference list is now extended, it was also request from other reviewers too.
2.Line 68: Is it necessary to describe the simplified vector in details to make readers know what it is like?
I believe it helps the understanding although the details are not in the right place, and I think the question also referred to this. Now the polygon vector regarding parts is moved to the "2D polygon-based road representation" section and it is clarified. Thank you for drawing attention to this matter
3.Line 86: A comma misses before 'a road'.
Corrected, thank you.
4.Line 92: Are you trying to express that 'False positive curb points may appear ...' instead of 'May false positive curb points appear ...'? And please explain what false positive curb points are and why  false positive voxels are never on the road surface.
Yes, it is now corrected in the revised manuscript. It is indeed a good idea to explain, done that, thank you!
I cite the added sentences:
Line..."False positive curb points may appear behind the actual curb. Curb points are the boundary voxels between the curb and the road. False positive curb points are created behind the curb, e.g., due the similar 3D characteristic of the voxels from various artifacts. The artifact can be for example a public bench which stands out from the sidewalk the same way as the curb stands out from the road. This results false identification. The final polygon is created between the road and first curb points, which means later curb points do not compromise the final result."
5.Line 101: Please add the coordinate system in Figure 2.
The coordinate system is added, figure 2 is also altered by the request of another reviewer. It also has a colorbar and a reference scale to understand it better.
6.Line 353: Is any data contained in the result of qualitative estimation? If so, please add a table to present the data like detection success rate and so on.
The qualitative estimation improvement was also requested from another reviewer. Now those improvements are included, thank you!
We added the following into the revised paper:
" The Jetson AGX Xavier has an 8 core ARM architecture, 64-bit Carmel CPU which has a 2.26GHz clock speed. During the in-vehicle tests, we used Ubuntu 18.04 operating system, which is a common Linux distribution. During the log-file-based analysis, we observed that the solution runs up to 2x faster than real-time, which means 40 Hz in our case. In this case, an Intel Core i9 16 core x64 architecture i9-7960X CPU was used at 2.80GHz clock speed. During the log-file-based analysis, a Windows 11 machine with WSL was used."
" We used another metric which was presented in [30] to quantify our results. To evaluate the algorithm a high-end processor i9-77940X @ 3.10GHz and 16 GB RAM. On this architecture the core of the algorithms, without the data acquisition and publication was able to run in average 0.014 seconds, which means more than 70 Hz (see figure 12). On a slightly slower CPU (i9-7960X @2.80GHz) but together with the data acquisition and publishing it was still over 40Hz. We calculated the running time of the algorithm proposed in the second part of the experiment, as shown in figure 12. To compare our results to a state-of-the art performance, in [30] the average running time was 0.036 seconds, this result was achieved on an older CPU (i7-4790) and the results were together with the data acquisition step. It is confident to say that both [30] and our method can perform in real-time."
Reviewer 2 Report
This paper is exploring road and sidewalk detection in urban scenarios by focusing on the height parameters obtained from a LIDAR sensor. The authors claim that the performance of the proposed solution is above 20 Hz for a real-time curb detection application.
The introduction is clear but requires numbers and comparisons from the literature. The detection frequency, GPU/CPU speed, also extracted/evaluated parameters other than the curbs are important to mention from the literature.
In the abstract, the authors claim that the output frequency is over 20, however, according to the paper only 20 Hz results are given for the experiment. Is it possible to use this approach with the same accuracy for higher on lower speed conditions?
The working range of the LIDAR is not discussed in the paper.
The distance of the LIDAR from the ground and its angle should be addressed.
Authors should mention the type (2D/3D) of the LIDAR
Mostly RTK-GPS output frequency is lower than 10 Hz, so the authors should add which model is used in the experiment and how did they achieve to correlate the frequencies.
In Figures 2 and 10 using a color bar will be helpful to understand the depth of the image.
There are two figures labeled with the same figure numbers.
In the result part giving numbers to compare this study with literature is important.
The references are very limited concerning the extensive research area on-road lane detection. Additionally, the references are not including very similar studies such as a study from MDPI “10.3390/electronics7110276”.
Author Response
Thank you for your review, they are good remarks and some of them were mentioned by other reviewers too, what (which) also emphasizes how important they are.
The introduction is clear but requires numbers and comparisons from the literature. The detection frequency, GPU/CPU speed, also extracted/evaluated parameters other than the curbs are important to mention from the literature.
Thank you, we mention the GPU/CPU speeds not only in literature review but also in results section. This is a valuable additional information to the readers.
We added the following into the revised paper....
In the abstract, the authors claim that the output frequency is over 20, however, according to the paper only 20 Hz results are given for the experiment. Is it possible to use this approach with the same accuracy for higher on lower speed conditions?
Good point, we mentioned the over 20 Hz results in the abstract but forget to explain and show the results. Now this forgetfulness is corrected, thank you for pointing it.
We added the following into the revised paper....
The working range of the LIDAR is not discussed in the paper.
It is now included in the revised manuscript. Citation from the revised paper:
Line 451 "We used the Ouster OS1 64 channel 3D LIDAR with a range of 120 meters."
Thank you!
The distance of the LIDAR from the ground and its angle should be addressed.
It is done in the revised version
Line 452 "The LIDAR was at the top of the vehicle, 1.54 meters above the ground without tilting."
Authors should mention the type (2D/3D) of the LIDAR
Now it is mentioned and also the Related work section the cited papers are listed as 2D/3D LIDAR solution.
Line 451 "We used the Ouster OS1 64 channel 3D LIDAR with a range of 120 meters."
Line 75 " LIDAR sensors provide raw 3D spatial information and can be organized as single layer LIDAR (also referred as 2D LIDAR or laser scanner) and multi-layer, or 3D LIDAR."
Thank you, this do accelerate readability.
Mostly RTK-GPS output frequency is lower than 10 Hz, so the authors should add which model is used in the experiment and how did they achieve to correlate the frequencies.
This information is added, thank you! Basically, we used an RTK-INS hybrid solution which runs on 20 Hz (the same frequency as the LIDAR) Citation from the revised paper:
Line 455: "At this experiment we used an RTK-INS (Novatel PwrPak7D GPS) which provides accurate position data at 20 Hz."
In Figures 2 and 10 using a color bar will be helpful to understand the depth of the image.
Figure 2 update was also a request from another Reviewer, this is done, thank you!
There are two figures labeled with the same figure numbers.
This was an edit mistake, ant now it is corrected. Thank you!
In the result part giving numbers to compare this study with literature is important.
We provided the requested numbers (deviation, mean and also the raw data) in a similar format than more cited papers, this can be used to compare.
Line 433: "We used another metric which was presented in [30] to quantify our results. To evaluate the algorithm a high-end processor i9-77940X @ 3.10GHz and 16 GB RAM. On this architecture the core of the algorithms, without the data acquisition and publication was able to run in average 0.014 seconds, which means more than 70 Hz (see figure 12). On a slightly slower CPU (i9-7960X @2.80GHz) but together with the data acquisition and publishing it was still over 40Hz. We calculated the running time of the algorithm proposed in the second part of the experiment, as shown in figure 12. To compare our results to a state-of-the art performance, in [30] the average running time was 0.036 seconds, this result was achieved on an older CPU (i7-4790) and the results were together with the data acquisition step. It is confident to say that both [30] and our method can perform in real-time. "
This is a very good advice, thank you!
The references are very limited concerning the extensive research area on-road lane detection. Additionally, the references are not including very similar studies such as a study from MDPI “10.3390/electronics7110276”.,
Extensive references are added now, also the mentioned paper. This is a similar approach, and we highlighted the similarities/differences between the works. Thank you once again!
Reviewer 3 Report
The manuscript Real-time LIDAR-based Urban Road and Sidewalk detection for Autonomous Vehicles is well written. It gives good insight on the road and sidewalk detection in urban scenarios is a challenge.
More references should be added,
The introduction should be elaborated. [R] Enhanced Performance of Fabry-Perot Tunable Filter by Groove Geometry Design of Double Folded Cantilever, Journal of Nanoelectronics and Optoelectronics J. Nanoelectron. Optoelectron. 15, 687–692 (2020) https://doi.org/10.1166/jno.2020.2830
There are many grammatical errors, that should be removed
How LIDAR results were validated?
Author Response
Thank you for your review. They are good remarks and some of them were mentioned by other reviewers too, what also emphasizes how important they are.
More references should be added,
This was a request form other Reviewers too; now extensive references are added. In our opinion the added references will help the reader to see a bigger picture, also the similarities/differences between the works are highlighted in more detail. Thank you!
Extensive references, which include also the mentioned paper, have been added. This is a similar approach, and we highlighted the similarities/differences between the works. Thank you once again!
The introduction should be elaborated. [R] Enhanced Performance of Fabry-Perot Tunable Filter by Groove Geometry Design of Double Folded Cantilever, Journal of Nanoelectronics and Optoelectronics J. Nanoelectron. Optoelectron. 15, 687–692 (2020) https://doi.org/10.1166/jno.2020.2830
Thanks for the suggestion! The subject of the mentioned paper is attractive and will certainly be considered in our future works, its is included in the references among some other new papers.
There are many grammatical errors, that should be removed
Thank you, we have revised the paper, now the new additions and modifications are highlighted with green. We and other reviewers did find many grammatical errors indeed, hopefully most of them are gone now.
Thank you for this suggestion it really enhanced the paper.
How LIDAR results were validated?
Thank you for asking this. Now we provided the requested (quantitative) validation (deviation, mean and also the raw data) in a similar format than more cited papers, this can be used to compare.
This is a very good question, it’s a pity that we did not include it in the first place. This is a very good observation and we include it in the actual version.
Reviewer 4 Report
Dear Authors,
Thanks so much for your manscript submission to MDPI Journal of Sensors. This paper presented a framework for road and sidewalk detection. The edge detection scheme combinede the Star-shaped search, X-zero and Z-zero methods in 3D voxels, then provides a polygon output by their approach which may be directly applied with a local trajectory planner. The authors specified that extensive real-time filed tests and off-line analysis from prior measurements and public datasets, have been performed. After my review, I think that this paper contains some interesting points in their experimental design and the mathematical modeling could be OK, while comprehensively, it requires some revisions. A few problematic issues on major and minor parts need to be addressed, which I specified as follows:
Major problematic issues to be addressed in the revised version:
a) Abstract: While the current version is very brief, it needs a major rewrite. It need to include methodology, technical approach, results (better some keynote quantitative remarks), and conclusions. Please consider reworking on this session and limit the length to ~150 words. Thanks a lot!
b) Introduction: suggested edits are listed in the following aspects: i) expand the review with both classical approaches and the state-of-the-arts, by summarizing some specific details (which should be concise and critical); ii) Add a short paragraph on the main summary of contributions from this set of work (in 2-3 manifolds, be specific), and make up the last paragragh to summarzie the organization on the remainder of this paper. Appreciate that.
c) Proposed solution: I noticed the formulation on mathematical models and the major steps of the proposed algorithm in contrast to Z-zero method, where the difference is sliding window-based approach. Besides this, is there any other innovative points that the authors can specify? What are the advantages of your approach when performing feature extraction?
d) Figures and Tables: First, Fig. 3 and Table 1 crossed over two pages, please arrange their locations decently in the updated version; the Fig. 10 in Page 12 should be named as Fig. 11. The related figures are visual results, no quantitative scores on proof. Hence, I suggest the authors to induce some performance metrics to present the validity and efficiency on their approach. It is better to shrink the size of Fig. 6 a little bit, and enhance the resolution of two subimages in Fig. 10. Besides, the size and resolution of each image, could be aligned to comply with MDPI template requirements.
e) Again, I think that as the authors stated, "qualitative estimation of our method" is not specific, if only Fig. 10 shows for that, other readers may doubt that this set of work is missing some keynote tests and lacking in the expected quantitative analysis. Please consider possible solutions to handle this problematic point, which is crucial.
f) Conclusion section: the title can be updated with "Conclusions and Future Work". Paragraphs 1-2 can be merged to a single paragraph as the main summary of your work, while two additional paragraphs on discussing the limitations of your study, the summary of research challenges and future research orientations, can be supplemented. It is great to share the source code and specify that in the last paragraph. I appreciate that favor.
g) References: the currently cited 20 references looks a bit weak. Please consider doing the following updates: i) Apply abbreviated formats on the title of each journal paper on cites (check an online template on Reference), supplement each with itatic style, information on volume and pages. ii) Make up the conference proceedings with missed information on time and locations; iii) Cite more state-of-the-art works which got published in Years 2019-2021, and uniformly comply with the wide-recognized formats in MDPI affilicated journals. For instance, Ref. [1] and [20] cite the same journal (Sensors), which the citation format is different, please fix that issue.
Minor problematic issues to be addressed in the revision procedure:
a) Do not hyphenate a word (which currently appears multiple time at the end of some lines to cross-over two adjacent lines). MS word file of MDPI online template has the options to adjust that. Thanks a lot!
b) Use of English should be improved in the updated version. The existing problems explicitly appears as follows: lacking in appropriate comma "," in some sentences (i.e, Line 23 and Line 110); the end of Line 66 at Page 2 missed a full stop. Similar issues can be fixed by careful proofreading. Meanwhile, some grammatical issues and mis-use of words should be calibrated in the revised version.
c) I suggest the authors downloading the MDPI online template (Sensors) to check the professional formats on title of second-level and third-level subsections as well as the formats on algorithms, then comply each of them.
Once again, thank you for your interests on publishing at MDPI Journal of Sensors. We expect you for the required edits which can be beneficial to improve the comprehensive quality of this paper. Best of luck to your future paper acceptance. Stay safe!
Best regards,
Yours sincerely,
Author Response
Thank you for your very detailed review. It was extremely helpful. Some of them were mentioned by other reviewers too, what also emphasizes how important they are. I think the correction based on the suggestion increased the readability and are a big help for the readers to compare and evaluate our results regard to other solutions. Thank you once again!
Major problematic issues to be addressed in the revised version:
- a) Abstract: While the current version is very brief, it needs a major rewrite. Itneed to include methodology, technical approach, results (better some keynote quantitative remarks), and conclusions. Please consider reworking on this session and limit the length to ~150 words. Thanks a lot!
The abstract is now revised, the new length is 144 words, methodology, technical approach and results are now highlighted in the abstract and also are emphasized inside the paper. This advice is really boosted readability, we really appreciate this suggestion!
- b) Introduction: suggested edits are listed in the following aspects:i) expand the review with both classical approaches and the state-of-the-arts, by summarizing some specific details (which should be concise and critical); ii) Add a short paragraph on the main summary of contributions from this set of work (in 2-3 manifolds, be specific), and make up the last paragragh to summarzie the organization on the remainder of this paper. Appreciate
The introduction part is now separated from the related work section, hopefully this helps the reader.
i) the review is now expanded, both on classical approaches and state-of-the-art (Years 2019-2021) results are in more detail, the cited literature is extended
ii) The introduction now contains the main summary of contributions, also the organization of the paper is now included
Thank you!
- c) Proposed solution: I noticed the formulation on mathematical models and the major steps of the proposed algorithm in contrast to Z-zero method, where the difference is sliding window-based approach. Besides this,is there any other innovative points that the authors can specify? What are the advantages of your approach when performing feature extraction?
The observation is correct in terms of: from the 3 detection sub-methods (star-shaped search, X-zero, Z-zero) star shaped and x-zero was completely original, but z-zero was inspired by Zhang’s and Dolan’s Road-Segmentation-Based Curb Detection Method. Actually, we even tried to contact Zhang about the algorithm, so we can compare ours to the original implementation. (C++ implementation can have an effect on the time performance, but of course not the working performance of the algorithm). John M. Dolan was kind to respond, and he suggested to contact Zang via email, but finally we couldn’t reach him. To summarize we had to implement our own version where we used this sliding window-based approach, which we found was advantageous. To answer your questions: our approach has the mentioned 3 sub-methods, all of them have the same purpose, but they work on a different principle. This adds robustness but from their parallel implementation does not seriously affect the time performance, all of the 3 methods enabled can run in real time. As an addition our method even produces a 2D polygon-based road representation, which is also a novelty. This can be directly used by a local planner, no need to transform it to occupancy grid mapping or any other map. Also, our approach is deterministic meaning to the same LIDAR input always the same output is produced. Many approaches do not work this way, which is not necessary a negative thing if it is used in research. Our approach as a deterministic method may suits better for automotive grade certification in the future. But at this part of the development, it is not a request.
The necessary explanation is also added to the revised version. Thank you for this suggestion!
- d) Figures and Tables: First, Fig. 3 and Table 1 crossed over two pages, please arrange their locations decently in the updated version;the Fig. 10 in Page 12 should be named as 11. The related figures are visual results, no quantitative scores on proof. Hence, I suggest the authors to induce some performance metrics to present the validity and efficiency on their approach. It is better to shrink the size of Fig. 6 a little bit, and enhance the resolution of two subimages in Fig. 10. Besides, the size and resolution of each image, could be aligned to comply with MDPI template requirements.
We have corrected the locations of the mentioned figures and table. With the major revision also the text flow was changed which and we adjusted the graphical elements accordance to this.
We attached the figures in vector format (pdf) and where it was not possible, we used printing quality 300 dpi or higher (png).
We used metric which was based on the work on Sun, X. Zhao, Z. Xu, R. Wang, H. Min A 3D LiDAR Data-Based Dedicated Road Boundary Detection Algorithm for Autonomous Vehicles, IEEE Access, 2019 to quantify our results. This lets us compare our results, although we had a newer CPU and our methodology does not include the data acquisition part only the core of the algorithm.
- e) Again, I think that as the authors stated, "qualitative estimation of our method" is not specific, if only Fig. 10 shows for that, other readers may doubt that this set of work is missing some keynote tests and lacking in the expected quantitative analysis. Please consider possible solutions to handle this problematic point, which is crucial.
We have addressed also this crucial remark, thank you for letting us know. Now quantitative results are present in the revised paper. Namely, we used a metric to evaluate the proposed method and verify if it can meet the real-time requirements. We conducted experiments. An example scenario is introduced in the paper where we drove for 75 seconds and recorded the LIDAR data at 20 Hz. Our experiments validated that the proposed method can perform in real-time both on a PC with an i9 CPU and an NVIDIA AGX Xavier embedded system We observed an average 0.014 seconds computation time for the core algorithm (on the i9). Even with the data acquisition and publishing part, it was still over 40Hz which is double the real-time performance.
Thank you for letting us know this issue.
We added the following into the revised paper:
" We used another metric which was presented in [30] to quantify our results. To evaluate the algorithm a high-end processor i9-77940X @ 3.10GHz and 16 GB RAM. On this architecture, the core of the algorithms, without the data acquisition and publication was able to run on average 0.014 seconds, which means more than 70 Hz (see figure 12). On a slightly slower CPU (i9-7960X @2.80GHz) but together with the data acquisition and publishing it was still over 40Hz. We calculated the running time of the algorithm proposed in the second part of the experiment, as shown in figure 12. To compare our results to a state-of-the art performance, in [30] the average running time was 0.036 seconds, this result was achieved on an older CPU (i7-4790) and the results were together with the data acquisition step. It is confident to say that both [30] and our method can perform in real-time.
"
- f) Conclusion section: the title can be updated with "Conclusions and Future Work". Paragraphs 1-2 can be merged to a single paragraph as the main summary of your work, while two additionalparagraphs on discussing the limitations of your study, the summary of research challenges and future research orientations, can be supplemented. It is great to share the source code and specify that in the last paragraph. I appreciate that favor.
The title is updated to "Conclusions and Future Work". Paragraphs 1-2 is merged now to a single paragraph as the main summary of our work. We extensively added the limitation of our work which was not addressed in detail before. Now more aspects of the limitations are added, thank you! _
We added the following into the revised paper:
" It is mentioned that the solution has limitations. Both the X-zero and the Z-zero algorithms require the LIDAR to be in a parallel position with respect to the road surface. Although this is a common sensor setup [34-39] and our vehicles are equipped this way, there are notable cases where it is recommended to set up it differently. For instance [38] has LIDARs both straight (parallel to the road) and tilted way. Also [40] only uses tilted LIDAR, so as an example our solution is unsuitable for this sensor setup.
A new type of LIDAR, called solid-state technology is getting higher interest in the scientific community [41-42]. These sensors have not yet been fully commercialized, they promise higher operational life and low power. They also produce structured 3D information but are organized differently. As a further limitation, the proposed method does not support this technology. The 3D data from the solid-state technology is not in the classical channel-based organization, thus our method cannot calculate the channel based sliding window, on which the later steps rely. As mentioned, there are several limitations regarding our method. Also concatenating two traditional spinning LIDAR data produces a similar result. Concatenating multiple LIDAR streams is quite common, thus this is a more serious limitation. The concatenated LIDAR data from the method point of view resembles to the solid-state data. The current method is not able to identify classic channel information.
Our approach is designed for autonomous vehicles, as a future work, it could be extended to environmental monitoring [43] or map generation [27]. Additional future work may consist of overcoming the mentioned limitations. "
- g) References: the currently cited 20 referenceslooks a bit weak. Please consider doing the following updates: i) Apply abbreviated formats on the title of each journal paper on cites (check an online template on Reference), supplement each with itatic style, information on volume and pages. ii) Make up the conference proceedings with missed information on time and locations; iii) Cite more state-of-the-art works which got published in Years 2019-2021, and uniformly comply with the wide-recognized formats in MDPI affilicated For instance, Ref. [1] and [20] cite the same journal (Sensors), which the citation format is different, please fix that issue.
Thank you, this is also corrected.
i) we supplemented journals with italic style, added information on volume and pages
ii) added missing time and locations the conference proceedings
iii) cited more state-of-the-art works which got published in 2019-2021
fixed the [1] and [20] problem
Minor problematic issues to be addressed in the revision procedure:
- a) Do not hyphenate a word (which currently appears multipletime at the end of some lines to cross-over two adjacent lines). MS word file of MDPI online template has the options to adjust that. Thanks a lot!
Done that, the paper has a cleaner look now. Thank you!
- b) Use of English should be improved in the updated version. The existing problems explicitlyappears as follows: lacking in appropriate comma "," in some sentences (e, Line 23 and Line 110); the end of Line 66 at Page 2 missed a full stop. Similar issues can be fixed by careful proofreading. Meanwhile, some grammatical issues and mis-use of words should be calibrated in the revised version.
Done this, thank you!
- c) I suggest the authors downloading the MDPI online template (Sensors)to check the professional formats on title of second-level and third-level subsections as well as the formats on algorithms, then comply each of them.
Thank you, we done that, now it is clear what was wrong from formatting point of view. The revised paper is fixed accordance the MDPI online Sensor's template.
Once again, thank you for your interests on publishing at MDPI Journal of Sensors. We expect you for the required edits which can be beneficial to improve the comprehensive quality of this paper. Best of luck to your future paper acceptance. Stay safe!
We all agree about that. We got extremely helpful requests and we hope that the revised version improved the comprehensive quality of the paper. We learned a lot from your remarks, hopefully the quality of the enhancement meets the requirement of the MPDI Sensors Journal. Thank you once again Stay safe too!
Round 2
Reviewer 1 Report
The manuscript is modified well and can be published. Congratulations!
Reviewer 2 Report
Thank you for addressing my comments